# REVISITING ACTIVATION FUNCTION DESIGN FOR IMPROVING ADVERSARIAL ROBUSTNESS AT SCALE

## ABSTRACT

Modern ConvNets typically use ReLU activation function. Recently smooth activation functions have been used to improve their accuracy. Here we study the role of smooth activation function from the perspective of adversarial robustness. We find that ReLU activation function significantly weakens adversarial training due to its non-smooth nature. Replacing ReLU with its smooth alternatives allows adversarial training to find harder adversarial training examples and to compute better gradient updates for network optimization.

We focus our study on the large-scale ImageNet dataset. On ResNet-50, switching from ReLU to the smooth activation function SILU improves adversarial robustness from 33.0% to 42.3%, while also improving accuracy by 0.9% on ImageNet. Smooth activation functions also scale well with larger networks: it helps EfficientNet-L1 to achieve 82.2% accuracy and 58.6% robustness, largely outperforming the previous state-of-the-art defense by 9.5% for accuracy and 11.6% for robustness. Models are available at `https://rb.gy/qt8jya`.

## 1 INTRODUCTION

It is known that convolutional neural networks can be easily fooled by adversarial examples (Szegedy et al., 2014). To improve robustness, many efforts have been made (Papernot et al., 2016; Guo et al., 2018; Xie et al., 2018; Liu et al., 2018; Pang et al., 2019; Schott et al., 2019); while adversarial training (Goodfellow et al., 2015; Kurakin et al., 2017; Madry et al., 2018), which trains networks with adversarial examples on-the-fly, stands as one of the most effective methods. Later studies further improve adversarial training by feeding networks with harder adversarial examples (Wang et al., 2019b), maximizing the margin of networks (Ding et al., 2020), optimizing a regularized surrogate loss (Zhang et al., 2019), *etc*. While these methods achieve stronger adversarial robustness, they sacrifice accuracy on clean inputs. It is generally believed such trade-off between accuracy and robustness might be inevitable (Tsipras et al., 2019), except for enlarging network capacities, *e.g*., making wider or deeper networks (Madry et al., 2018; Xie & Yuille, 2020).

Another popular direction for increasing robustness against adversarial attacks is gradient masking (Papernot et al., 2017; Xie et al., 2018; Samangouei et al., 2018; Song et al., 2018; Ma et al., 2018; Guo et al., 2018). With the degenerated gradient quality, attackers cannot successfully optimize the targeted loss and therefore fail to circumvent such defenses. Nonetheless, the gradient masking operation will be ineffective to offer robustness if its differentiable approximation is used for generating adversarial examples (Athalye et al., 2018).

To effectively build robust models, we hereby rethink the relationship between gradient quality and adversarial robustness, especially in the context of adversarial training where gradients are applied more frequently than standard training. In addition to computing gradients to update network parameters, adversarial training also requires gradient computation for generating training samples. Guided by this principle, we identify ReLU, a widely-used activation function in modern ConvNets, significantly weakens adversarial training due to its non-smooth nature, *e.g*., ReLU's gradient gets an abrupt change (from 0 to 1) when its input is close to zero (see Figure 1).

In this paper, we revisit the activation function design for improving adversarial robustness, with a special focus on the large-scale ImageNet dataset (Russakovsky et al., 2015). To fix the issue

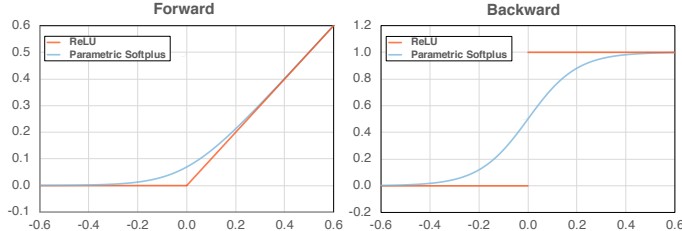

**Figure 1:** The visualization of ReLU and Parametric Softplus. *Left panel*: the forward pass for ReLU (blue curve) and Parametric Softplus (red curve). *Right panel*: the first derivatives for ReLU (blue curve) and Parametric Softplus (red curve). Different from ReLU, Parametric Softplus is smooth with continuous derivatives.

induced by ReLU as aforementioned, we propose to apply its smooth approximations[1] for improving the gradient quality in adversarial training (Figure 1 shows Parametric Softplus, an example of smooth approximations for ReLU). Our experiment results show that switching from ReLU to its smooth approximations in adversarial training can substantially improves adversarial robustness. For instance, by training with the computationally cheap *single-step PGD attacker*[2] on ImageNet, the smooth activation function SILU significantly improves the robustness of the ResNet-50 baseline (which uses ReLU) by 9.3%, from 33.0% to 42.3%, meanwhile increasing standard accuracy by 0.9%. In addition, we note this performance improvement in both robustness and accuracy comes for "free", as the change of activation function does not incur additional computational cost.

We next explore the limits of adversarial training with smooth activation function using larger networks. We obtain the best result by using EfficientNet-L1 (Tan & Le, 2019; Xie et al., 2020), which achieves 82.2% accuracy and 58.6% robustness on ImageNet, significantly outperforming the previous state-of-the-art defense (Qin et al., 2019) by 9.5% for accuracy and 11.6% for robustness.

## 2   RELATED WORKS

**Adversarial training.** Adversarial training improves robustness by training models on adversarial examples (Szegedy et al., 2014; Goodfellow et al., 2015; Kurakin et al., 2017; Madry et al., 2018). Existing works suggest that, to further adversarial robustness, we need to either sacrifice accuracy on clean inputs (Wang et al., 2019b; 2020; Zhang et al., 2019; Ding et al., 2020), or incur additional computational cost (Madry et al., 2018; Xie & Yuille, 2020; Xie & et al., 2019). This phenomenon is referred to as *no free lunch in adversarial robustness* (Tsipras et al., 2019; Nakkiran, 2019; Su et al., 2018). In this paper, we show that, using smooth activation function in adversarial training, adversarial robustness can be improved for "free"—no accuracy degradation on clean images and no additional computational cost incurred.

Our work is also related to the theoretical study (Sinha et al., 2018), which shows replacing ReLU with smooth alternatives can help networks get a tractable bound when certifying distributional robustness. In this paper, we empirically corroborate the benefits of utilizing smooth activations is also observable in the practical adversarial training on the real-world dataset using large networks.

**Gradient masking.** Besides training models on adversarial data, other ways for improving adversarial robustness include defensive distillation (Papernot et al., 2016), gradient discretization (Buckman et al., 2018; Rozsa & Boult, 2019; Xiao et al., 2019), dynamic network architectures (Dhillon et al., 2018; Wang et al., 2018; 2019a; Liu et al., 2018; Lee et al., 2020; Luo et al., 2020), randomized transformations (Xie et al., 2018; Bhagoji et al., 2018; Xiao & Zheng, 2020; Raff et al., 2019; Kettunen et al., 2019; AprilPyone & Kiya, 2020), adversarial input denoising/purification (Guo et al., 2018; Prakash et al., 2018; Meng & Chen, 2017; Song et al., 2018; Samangouei et al., 2018; Liao et al., 2018; Bhagoji et al., 2018; Pang et al., 2020; Xu et al., 2017; Dziugaite et al., 2016), *etc*. Nonetheless, most of these defense methods degenerate the gradient quality of the protected models, therefore could induce the gradient masking issue (Papernot et al., 2017). As argued in (Athalye et al., 2018), defense methods with gradient masking may offer a false sense of adversarial robustness. In contrast to these works, we aim to improve adversarial robustness by providing networks with better gradients, but in the context of adversarial training.

---

[1] The term "smooth" hereby refer to this function satisfies the property of being $\mathcal{C}^1$ smooth, *i.e.*, its first derivative is continuous everywhere.

[2] In practice, we note that single-step PGD adversarial training is only ~1.5× slower than standard training.

**Table 1: ReLU significantly weakens adversarial training**. By improving gradient quality for either the adversarial attacker or the network optimizer, resulted models obtains better robustness than the ReLU baseline. The best robustness is achieved by adopting better gradients for both the attacker and the network optimizer. *Note that, for all these model, ReLU is always used at their forward pass, albeit their backward pass may get probed in this ablation.*

| Network | Improving Gradient Quality for the Adversarial Attacker | Improving Gradient Quality for the Network Optimizer | Accuracy (%) | Robustness (%) |
|---|---|---|---|---|
| ResNet-50 | ✗ | ✗ | 68.8 | 33.0 |
| | ✓ | ✗ | 68.3 (-0.5) | 34.5 (+1.5) |
| | ✗ | ✓ | 69.4 (+0.6) | 35.8 (+2.8) |
| | ✓ | ✓ | 68.9 (+0.1) | 36.9 (+3.9) |

# 3 ReLU Weakens Adversarial Training

In this section, we perform a series of controlled experiments, *specifically in the backward pass of gradient computations*, to investigate how ReLU weakens, and how its smooth approximation strengthens adversarial training.

## 3.1 Adversarial Training

Adversarial training (Szegedy et al., 2014; Goodfellow et al., 2015; Madry et al., 2018), which trains networks with adversarial examples on-the-fly, aims to optimize the following framework:

$$\arg\min_{\theta} \mathbb{E}_{(x,y)\sim\mathbb{D}} \left[ \max_{\epsilon\in\mathbb{S}} L(\theta, x + \epsilon, y) \right], \tag{1}$$

where $\mathbb{D}$ is the underlying data distribution, $L(\cdot, \cdot, \cdot)$ is the loss function, $\theta$ is the network parameter, $x$ is a training sample with the ground-truth label $y$, $\epsilon$ is the added adversarial perturbation, and $\mathbb{S}$ is the allowed perturbation range. To ensure the generated adversarial perturbation $\epsilon$ is human-imperceptible, we usually restrict the perturbation range $\mathbb{S}$ to be small (Szegedy et al., 2014; Goodfellow et al., 2015; Lou et al., 2015).

As shown in equation 1, adversarial training consists of two computation steps: an *inner maximization step*, which computes adversarial examples, and an *outer minimization step*, which computes parameter updates.

**Adversarial training setup.** We choose ResNet-50 (He et al., 2016) as the backbone network, where ReLU is used by default. We apply PGD attacker (Madry et al., 2018) to generate adversarial perturbations $\epsilon$. Specifically, we select the cheapest version of PGD attacker, *single-step PGD* (PGD-1), to lower the training cost. Following (Shafahi et al., 2019; Wong et al., 2020), we set the maximum per-pixel change $\epsilon = 4$ and the attack step size $\beta = 4$ in PGD-1. We follow the standard ResNet training recipes to train models on ImageNet: models are trained for a total of 100 epochs using momentum SGD optimizer, with the learning rate decreased by $10\times$ at the 30-th, 60-th and 90-th epoch; no regularization except a weight decay of 1e-4 is applied.

When evaluating adversarial robustness, we measure the model's top-1 accuracy against the 200-step PGD attacker (PGD-200) on the ImageNet validation set, with the maximum perturbation size $\epsilon = 4$ and the step size $\beta = 1$. We note 200 attack iterations are enough to let PGD attacker converge. Meanwhile, we report the model's top-1 accuracy on the original ImageNet validation set.

## 3.2 How Does Gradient Quality Affect Adversarial Training?

Figure 1 shows ReLU is non-smooth. ReLU's gradient takes an abrupt change, when its input is close 0, therefore significantly degrades the gradient quality. We conjecture this non-smooth nature hurts the training process, especially when we train models adversarially. This is because, compared to standard training which only computes gradients for updating network parameter $\theta$, adversarial training requires additional computations for the inner maximization step to craft the perturbation $\epsilon$.

To fix this problem, we first introduce a smooth approximation of ReLU, named *Parametric Softplus* (Nair & Hinton, 2010), as $f(\alpha, x) = \frac{1}{\alpha}\log(1 + \exp(\alpha x))$, where the hyperparameter $\alpha$ is used to control the curve shape. Its derivative w.r.t. the input $x$ is:

$$\frac{d}{dx}f(\alpha, x) = \frac{1}{1 + \exp(-\alpha x)} \tag{2}$$

To better approximate the curve of ReLU, we empirically set $\alpha = 10$. As shown in Figure 1, compared to ReLU, Parametric Softplus ($\alpha=10$) is smooth because it has a continuous derivative.

With Parametric Softplus, we next diagnose how gradient quality in *the inner maximization step* and *the outer minimization step* affects accuracy and robustness of ResNet-50 in adversarial training. *To clearly benchmark the effects, we only substitute ReLU with equation 2 in the backward pass, while leaving the forward pass unchanged*, i.e., ReLU is always used for model inference.

**Improving gradient quality for the adversarial attacker.** We first take a look at the effects of gradient quality on computing adversarial examples (*i.e.*, *the inner maximization step*) during training. More precisely, in the inner step of adversarial training, we use ReLU in the forward pass, but Parametric Softplus in the backward pass; and in the outer step of adversarial training, we use ReLU in both the forward and the backward pass.

As shown in the second row of Table 1, when the attacker uses Parametric Softplus's gradient (equation 2) to craft adversrial training samples, the resulted model exhibits a performance trade-off. Compared to the ReLU baseline, it improves adversarial robustness by 1.5% but degrades standard accuracy by 0.5%.

Interestingly, we note such performance trade-off can also be observed if harder adversarial examples are used in adversarial training (Wang et al., 2019b). This observation motivates us to hypothesize that better gradients for the inner maximization step may boost the attacker's strength during training. To verify this hypothesis, we evaluate the robustness of two ResNet-50 models via PGD-1 (*vs*. PGD-200 in Table 1), one with standard training and one with adversarial training. Specifically, during the evaluation, the PGD-1 attacker uses ReLU in the forward pass, but Parametric Softplus in the backward pass. With better gradients, PGD-1 attacker is strengthened and hurts models more: it can further decrease the top-1 accuracy by 4.0% (from 16.9% to 12.9%) on the ResNet-50 with standard training, and by 0.7% (from 48.7% to 48.0%) on the ResNet-50 with adversarial training (both results are not shown in Table 1).

**Improving gradient quality for network parameter updates.** We then study the role of gradient quality on updating network parameters (*i.e.*, *the outer minimization step*) during training. More precisely, in the inner step of adversarial training, we use ReLU; but in the outer step of adversarial training, we use ReLU in the forward pass, and Parametric Softplus in the backward pass.

Surprisingly, by setting the network optimizer to use Parametric Softplus's gradient (*i.e.*, equation 2) to update network parameters, this strategy improves adversarial robustness for "free". As shown in the third row of Table 1, without incurring additional computations, adversarial robustness is boosted by 2.8%, and meanwhile accuracy is improved by 0.6%, compared to the ReLU baseline. We note the corresponding training loss also gets lower: the cross-entropy loss on the training set is reduced from 2.71 to 2.59. These results of better robustness and accuracy, and lower training loss together suggest that, by using ReLU's smooth approximation in the backward pass of the outer minimization step, networks are able to compute better gradient updates in adversarial training.

Interestingly, we observe that better gradient updates can also improve standard training. For example, with ResNet-50, training with better gradients can improve accuracy from 76.8% to 77.0%, and reduces the corresponding training loss from 1.22 to 1.18. These results on both adversarial training and standard training suggest updating network parameters using better gradients could serve as a principle for improving performance in general, while keeping the inference process of the model unchanged (*i.e.*, ReLU is always used for inference).

**Improving gradient quality for both the adversarial attacker and network parameter updates.** Given the observation that improving ReLU's gradient for either the adversarial attacker or the network optimizer benefits robustness, we further enhance adversarial training by replacing ReLU with Parametric Softplus in all backward passes, but keeping ReLU in all forward passes.

As expected, such a trained model reports the best robustness so far. As shown in the last row of Table 1, it substantially outperforms the ReLU baseline by 3.9% for robustness. Interestingly, this improvement still comes for "free"—it reports 0.1% higher accuracy than the ReLU baseline. We conjecture this is due to the positive effect on accuracy brought by computing better gradient updates (increase accuracy) slightly overrides the negative effects on accuracy brought by creating harder training samples (hurt accuracy) in this experiment.

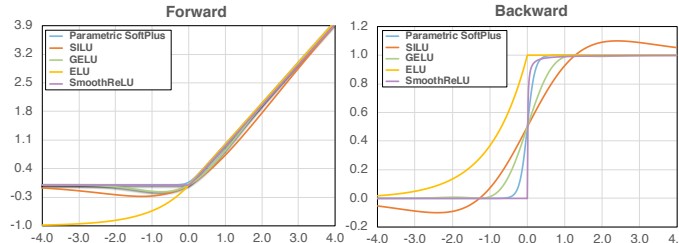

**Figure 2:** Visualizations of 5 different smooth activation functions and their derivatives.

### 3.3 CAN RELU'S GRADIENT ISSUE BE REMEDIED?

**More attack iterations.** It is known that increasing the number of attack iterations can create harder adversarial examples (Madry et al., 2018; Dong et al., 2018). We confirm in our own experiments that by training with PGD attacker with more iterations, the resulted model exhibits a similar behavior to the case where we apply better gradients for the attacker. For example, by increasing the attacker's cost by $2\times$, PGD-2 improves the ReLU baseline by 0.6% for robustness while losing 0.1% for accuracy. This result suggests *we can remedy ReLU's gradient issue in the inner step of adversarial training if more computations are given*.

**Training longer.** It is also known that longer training can lower the training loss (Hoffer et al., 2017), which we explore next. Interestingly, by extending the default setup to a $2\times$ training schedule (*i.e.*, 200 training epochs), though the final model indeed achieves a lower training loss (from 2.71 to 2.62), there still exhibits a performance trade-off between accuracy and robustness. Longer training gains 2.6% for accuracy but loses 1.8% for robustness. On the contrary, our previous experiment shows that applying better gradients to optimize networks improves both robustness and accuracy. This discouraging result suggests that *training longer cannot fix the issues in the outer step of adversarial training caused by ReLU's poor gradient*.

**Conclusion.** Given these results, we conclude that ReLU significantly weakens adversarial training. Moreover, it seems that the degenerated performance cannot be simply remedied even with training enhancements (*i.e.*, increasing the number of attack iterations & training longer). We identify that the key is ReLU's poor gradient—by replacing ReLU with its smooth approximation *only in the backward pass* substantially improves robustness, even without sacrificing accuracy and incurring additional computational cost. In the next section, we show that making activation functions smooth is a good design principle for enhancing adversarial training in general.

## 4 ADVERSARIAL TRAINING WITH SMOOTH ACTIVATION FUNCTIONS

As demonstrated in Section 3, improving ReLU's gradient can both strengthen the attacker and provide better gradient updates in adversarial training. Nonetheless, this strategy may be suboptimal as *there still exists a discrepancy between the forward pass (for which we use ReLU) and the backward pass (for which we use Parametric Softplus) when training the networks*.

To fully exploit the potential of training with better gradients, we hereby propose to exclusively apply smooth actvaition functions (*in both the forward pass and the backward pass*) in adversarial training. Noe that we keep all other network components exactly the same, as most of them are smooth and will not result in the issue of poor gradient.[3]

### 4.1 SMOOTH ACTIVATION FUNCTIONS

We consider the following activation functions as ReLU's smooth approximations in adversarial training (Figure 2 plots these functions and their derivatives):

- **Softplus** (Nair & Hinton, 2010): $\text{Softplus}(x) = \log(1 + \exp(x))$. We also consider its parametric version where $\alpha$ is set to 10 as in Section 3.

---

[3]We hereby ignore the gradient issue caused by max pooling, which is also non-smooth. This is because modern architectures rarely adopt it, *e.g.* only one max pooling layer is adopted in ResNet (He et al., 2016), and none is adopted in EfficientNet (Tan & Le, 2019).

- **SILU** (Elfwing et al., 2018; Hendrycks & Gimpel, 2016; Ramachandran et al., 2017): $\text{SILU}(x) = x \cdot \text{sigmoid}(x)$. Compared to other activation functions, SILU has a non-monotonic "bump" when $x < 0$.

- **Exponential Linear Unit** (ELU) (Clevert et al., 2016): if $x \geq 0$, $\text{ELU}(x, \alpha) = x$; otherwise $\text{ELU}(x, \alpha) = \alpha(\exp(x) - 1)$, $\text{ELU}(x, \alpha) = x$ if $x \geq 0$, $\alpha(\exp(x) - 1)$ otherwise, where we set $\alpha = 1$ as default. Note that when $\alpha \neq 1$, the gradient of ELU is not continuously differentiable anymore. We will discuss the effects of these non-smooth variants of ELU ($\alpha \neq 1$) on adversarial training in Section 4.3.

- **Gaussian Error Linear Unit** (GELU) (Hendrycks & Gimpel, 2016): $\text{GELU}(x) = x \cdot \Phi(x)$, where $\Phi(x)$ is cumulative distribution function of the standard normal distribution.

**Main results.** We follow the settings in Section 3 to adversarially train ResNet-50 equipped with smooth activation functions. The results are shown in Figure 3. Compared to the ReLU baseline, all smooth activation functions substantially boost robustness, while keeping standard accuracy almost the same. For example, smooth activation functions at least boost robustness by 5.7% (using Parametric Softplus, from 33% to 38.7%). Our strongest robustness is achieved by SILU, which enables ResNet-50 to achieve 42.3% robustness and 69.7% standard accuracy. We believe these results can be furthered if more advanced smooth alternatives (*e.g.*, (Misra, 2019; Lokhande et al., 2020; Biswas et al., 2020)) are used.

We then compare to the setting in Section 3 where Parametric Softplus is only applied at the backward pass during training. Interestingly, by additionally replacing ReLU with Parametric Softplus at the forward pass, the resulted model further improves robustness by 1.8% (from 36.9% to 38.7%) while without hurting accuracy. This result demonstrates the importance of applying smooth activation functions in both forward and backward passes in adversarial training.

### 4.2 RULING OUT THE EFFECT FROM $x < 0$

Compared to ReLU, in addition to being smooth, the functions above have non-zero responses to negative inputs ($x < 0$) which may also affect adversarial training. To rule out this factor, inspired by the design of Rectified Smooth Continuous Unit in (Shamir et al., 2020), we hereby propose SmoothReLU, which flattens the activation function by only modifying ReLU after $x \geq 0$,

$$\text{SmoothReLU}(x, \alpha) = \begin{cases} x - \frac{1}{\alpha} \log(\alpha x + 1) & \text{if } x \geq 0, \\ 0 & \text{otherwise,} \end{cases} \quad (3)$$

where $\alpha$ is a learnable variable shared by all channels of one layer, and is constrained to be positive. We note SmoothReLU is always continuously differentiable regardless the value of $\alpha$,

$$\frac{d}{dx} \text{SmoothReLU}(x, \alpha) = \begin{cases} \frac{\alpha x}{1 + \alpha x} & \text{if } x \geq 0, \\ 0 & \text{otherwise.} \end{cases} \quad (4)$$

Note SmoothReLU converges to ReLU when $\alpha \to \infty$. Additionally, in practice, the learnable parameter $\alpha$ needs to be initialized at a large enough value (*e.g.*, 400 in our experiments) to avoid the gradient vanishing problem at the beginning of training. We plot SmoothReLU and its first derivative in Figure 2.

We observe SmoothReLU substantially outperforms ReLU by 7.3% for robustness (from 33.0% to 40.3%), and by 0.6% for accuracy (from 68.8% to 69.4%), therefore clearly demonstrates the importance of a function to be smooth, and rules out the effect from having responses when $x < 0$.

### 4.3 STABILIZING ADVERSARIAL TRAINING WITH ELU USING CELU

In the analysis above, we show that adversarial training can be greatly improved by replacing ReLU with its smooth approximations. To further demonstrate the importance of being smooth, we provide another case study by using ELU (Clevert et al., 2016). The first derivative of ELU is shown below:

$$\frac{d}{dx} \text{ELU}(x, \alpha) = \begin{cases} 1 & \text{if } x \geq 0, \\ \alpha \exp(x) & \text{otherwise.} \end{cases} \quad (5)$$

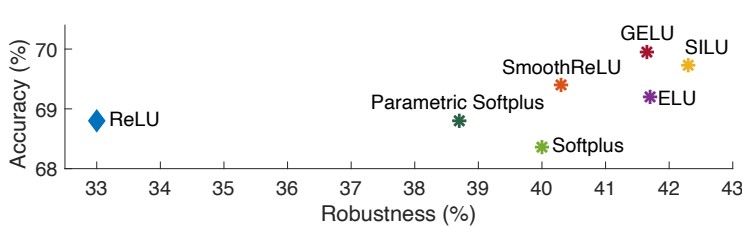

| $\alpha$ | Robustness (%) | |
|---|---|---|
| | ELU | CELU |
| 1 | 41.1 | |
| 1.2 | -0.3 | +0.1 |
| 1.4 | -2.0 | -0.3 |
| 1.6 | -3.7 | -0.3 |
| 1.8 | -6.2 | -0.2 |
| 2.0 | -7.9 | -0.5 |

**Figure 3:** Smooth activation functions improve adversarial training. Compared to ReLU, all smooth activation functions significantly boost robustness, while keeping accuracy almost the same.

**Table 2:** Robustness comparison between ELU (non-smooth when $\alpha \neq 1$) and CELU (always smooth $\forall \alpha$).

Here we mainly discuss the scenario when ELU is non-smooth, *i.e.*, $\alpha \neq 1$. As can be seen from equation 5, ELU's gradient is not continuously differentiable anymore, *i.e.*, $\alpha \exp(x) \neq 1$ when $x = 0$, therefore resulting in an abrupt gradient change like ReLU. Specifically, we consider the range $1.0 < \alpha \leq 2.0$, where the gradient abruption becomes more drastic with a larger value of $\alpha$.

We show adversarial training results in Table 2. Interestingly, we observe that adversarial robustness is highly dependent on the value of $\alpha$—the strongest robustness is achieved when the function is smooth (*i.e.*, $\alpha = 1.0$, 41.4% robustness), and all other choices of $\alpha$ monotonically decrease the robustness when $\alpha$ gradually approaches 2.0. For instance, with $\alpha = 2.0$, the robustness drops to only 33.2%, which is 7.9% lower than that of using $\alpha = 1.0$. The observed phenomenon here is consistent with our previous conclusion on ReLU—non-smooth activation functions significantly weaken adversarial training.

To stabilize adversarial training with ELU, we apply its smooth version, CELU (Barron, 2017), which re-parametrize ELU to the following format:

$$\mathrm{CELU}(x, \alpha) = \begin{cases} x & \text{if } x \geq 0, \\ \alpha \left( \exp \left( \frac{x}{\alpha} \right) - 1 \right) & \text{otherwise.} \end{cases} \quad (6)$$

Note that CELU is equivalent to ELU when $\alpha = 1.0$. The first derivatives of CELU can be written as follows:

$$\frac{d}{dx} \mathrm{CELU}(x, \alpha) = \begin{cases} 1 & \text{if } x \geq 0, \\ \exp \frac{x}{\alpha} & \text{otherwise.} \end{cases} \quad (7)$$

With this parameterization, CELU is now continuously differentiable regardless of the choice of $\alpha$. As shown in Table 2, we can observe that CELU greatly stabilizes adversarial training. Compared to the best result reported by ELU/CELU ($\alpha = 1.0$), the worst case in CELU ($\alpha = 2.0$) is merely 0.5% lower. Recall that this gap for ELU is 7.9%. This case study further support that it is important to apply smooth activation functions in adversarial training.

# 5 EXPLORING THE LIMITS OF ADVERSARIAL TRAINING WITH SMOOTH ACTIVATION FUNCTIONS

Recent works (Xie & Yuille, 2020; Gao et al., 2019) show that, compared to standard training, adversarial training exhibits a much stronger requirement for larger networks to obtain better performance. Nonetheless, previous explorations in this direction only consider either deeper networks (Xie & Yuille, 2020) or wider networks (Madry et al., 2018), which might be insufficient. To this end, we hereby present a systematic and comprehensive study on showing how network scaling up behaves in adversarial training, but using smooth activation functions. We set SILU as the default activation function, as it achieves the best robustness among all other candidates (see Figure 3).

## 5.1 SCALING-UP RESNET

We first perform the network scaling-up experiments with the ResNet family. In standard training, Tan *et al*. (Tan & Le, 2019) already show that, all three scaling-up factors, *i.e.*, *depth, width and image resolutions*, are important to further improve ResNet performance. We hereby examine the effects of these scaling-up factors in adversarial training with smooth activation functions. We choose ResNet-50 (with the default image resolution at 224) as the baseline network.

**Table 3:** Scaling-up ResNet with the smooth activation function SILU in adversarial training. We observe larger networks consistently get better performance.

| | Accuracy (%) | Robustness (%) |
|---|---|---|
| ResNet-50 | 69.7 | 42.3 |
| + 2x deeper (ResNet-101) | 72.9 (+3.2) | 45.5 (+3.2) |
| + 3x deeper (ResNet-152) | 73.9 (+4.2) | 46.0 (+3.7) |
| + 2x wider (ResNeXt-50-32x4d) | 71.2 (+1.5) | 42.5 (+0.2) |
| + 4x wider (ResNeXt-50-32x8d) | 73.6 (+3.9) | 45.1 (+2.8) |
| + image size 299 | 70.9 (+1.2) | 43.8 (+1.5) |
| + image size 380 | 71.6 (+1.9) | 44.1 (+1.8) |
| + 3x deeper & 4x wider (ResNeXt-152-32x8d) & image size 380 | **78.2 (+8.5)** | **51.2 (+8.9)** |

**Depth & width.** Previous works have shown that making networks deeper or wider can yield better model performance in standard adversarial training. We re-verify this conclusion using ResNet with the smooth activation function SILU. As shown in the second to fifth rows of Table 3, we confirm that both deeper or wider networks consistently outperform the baseline network. For instance, by training a $3\times$ deeper ResNet-152, it improves ResNet-50's performance by 4.2% for accuracy and 3.7% for robustness. Similarly, by training a $4\times$ wider ResNeXt-50-32x8d (Xie et al., 2017), it improves accuracy by 3.9% and robustness by 2.8%.

**Image resolution.** Though larger image resolution benefits standard training, it is generally believed that scaling up this factor will induce weaker adversarial robustness, as the attacker will have a larger room for crafting adversarial perturbations (Galloway et al., 2019). However, surprisingly, this belief could be invalid when taking adversarial training into consideration. As shown in the sixth and seventh rows of Table 3, ResNet-50 with the smooth activation function SILU consistently achieves better performance when training with larger image resolutions.

We conjecture this improvement is possibly due to larger image resolution (1) enables attackers to create stronger adversarial examples (Galloway et al., 2019) (which will hurt accuracy but improve robustness); and (2) increases network capacity for better representation learning (Tan & Le, 2019) (which will improve both accuracy and robustness); and the mixture of these two effects empirically yields a positive signal here.

**Compound scaling.** So far, we have confirmed that the basic scaling of depth, width and image resolution are all important scaling-up factors. As argued in (Tan & Le, 2019) for standard training, scaling up all these factors simultaneously will produce a much stronger model than just focusing on scaling up a single dimension. Hence we make an attempt to create a simple compound scaling for ResNet. As shown in the last row of Table 3, we can observe that the resulted model, ResNeXt-152-32x8d with input resolution at 380, achieves significantly better performance than the ResNet-50 baseline, *i.e.*, +8.5% for accuracy and +8.9% for robustness.

**Adversarial training with ReLU.** We first verify that the basic scaling of depth, width and image resolution also matter when ReLU is used in adversarial training *e.g.*, by scaling up ResNet-50 (33.0% robustness), the deeper ResNet-152 achieves 39.4% robustness (+6.4%), the wider ResNeXt-50-32x8d achieves 36.7% robustness (+3.7%), and the ResNet-50 with larger image resolution at 380 achieves 36.9% robustness (+3.9%). Nonetheless, all these robustness performances are still lower than the robustness achieved by the basic ResNet-50 with the smooth activation function SILU (42.3% robustness, first row of Table 3).

We also find compound scaling is more effective than basic scaling for adversarial training with ReLU, *e.g.*, ResNeXt-152-32x8d with input resolution at 380 reports 46.3% robustness. Although this result is much better than the basic scaling above, it is still ~5% lower than ResNet + SILU with compound scaling, *i.e.*, 46.3% v.s. 51.2%. In other words, even with larger networks, applying smooth activation functions remains essential for improving performance.

## 5.2 EFFICIENTNET RESULTS

The results on ResNet show scaling up networks with smooth activation functions in adversarial training effectively improves performance. Nonetheless, the applied scaling policies could be suboptimal, as they are hand-designed without any optimizations. EfficientNet (Tan & Le, 2019), which uses neural architecture search (Zoph & Le, 2016) to automatically discover the optimal factors for network compound scaling, provides a strong family of models for image recognition. Therefore we next use EfficientNet to replace ResNet. Note that all other training setups (*e.g.*, using SILU) are the same as described in our ResNet experiments.

Similar to ResNet, Figure 4 shows stronger EfficientNet consistently achieves better performance. For instance, by scaling the network from EfficientNet-B0 to EfficientNet-B7, the robustness is improved from 37.6% to 57.0%, and the accuracy is improved from 65.1% to 79.8%. Surprisingly, the improvement is still observable for larger networks: EfficientNet-L1 (Xie et al., 2020) further improves robustness by 1.0% and accuracy by 0.7% over EfficientNet-B7. All these results corroborate the finding in (Cubuk et al., 2018) that stronger backbone tend to yield higher robustness.

**Training enhancements.** So far all of our experiments follow the training recipes from ResNet, which may not be optimal for EfficientNet training. Therefore, as suggested in the original EfficientNet paper (Tan & Le, 2019), we adopt the following training setups in our experiments: we change weight decay from 1e-4 to 1e-5, and add Dropout (Srivastava et al., 2014), Stochastic Depth (Huang et al., 2016) and AutoAugment (Cubuk et al., 2019) to regularize the training process. Besides, we train models with longer schedule (*i.e.*, 200 training epochs) to better cope with these training enhancements, adopt the early stopping strategy to prevent the catastrophic overfitting issue in robustness (Wong et al., 2020), and save checkpoints using model weight averaging (Izmailov et al., 2018) to approximate the model ensembling for stronger robustness (Pang et al., 2019; Strauss et al., 2017). Additionally, as shown in the concurrent work (Rebuffi et al., 2020), applying weight averaging is the key for enabling data augmentation to improve robustness. With these training enhancements, our EfficientNet-L1 gets further improved, *i.e.*, +1.7% for accuracy (from 80.5% to 82.2%) and +0.6% for robustness (from 58.0% to 58.6%).

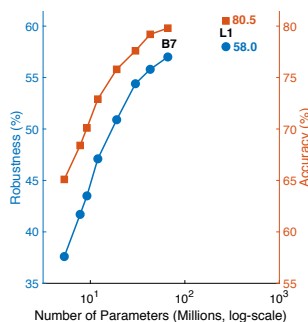

**Figure 4:** Scaling-up EfficientNet with the smooth activation function SILU in adversarial training. Note EfficientNet-L1 is not connected to the rest of the graph because it was not part of the compound scaling suggested by Tan *et al.* (Tan & Le, 2019).

**Comparing to the prior art (Qin et al., 2019).** Table 4 compares our best results with the prior art. By applying the smooth activation function in adversarial training, we are able to train a model with strong performance on both adversarial robustness and standard accuracy—our best model (*i.e.*, EfficientNet-L1) achieves 82.2% standard accuracy and 58.6% robustness, which largely outperforms the prior art (Qin et al., 2019) by 9.5% for standard accuracy and 11.6% for adversarial robustness. Note this improvement mainly stems from the facts that we hereby exploit a better activation function (SILU *vs*. Softplus) and a stronger architecture (EfficientNet-L1 *vs*. ResNet-152).

**Accuracy drop *vs*. model scale.** Finally, we emphasize a large reduction in the accuracy gap between adversarially trained models and stan-

**Table 4:** Comparison to the previous state-of-the-art.

|  | Accuracy (%) | Robustness (%) |
|---|---|---|
| Prior art (Qin et al., 2019) | 72.7 | 47.0 |
| EfficientNet (ours) | **82.2 (+9.5)** | **58.6 (+11.6)** |

dard trained models for large networks. With the training setup above, EfficientNet-L1 is able to attain 84.1% accuracy in standard training, while this accuracy only slightly decreases to 82.2% (-1.9%) in adversarial training. Note this gap is substantially smaller than the gap in ResNet-50 of 7.1% (76.8% in standard training v.s. 69.7% in adversarial training with the smooth activation function SILU). Moreover, it is also worth mentioning the high accuracy of 82.2% provides strong support to (Ilyas et al., 2019) on arguing robust features indeed can generalize well to clean inputs.

## 6 CONCLUSION

In this paper, we revisit the activation function design for improving adversarial robustness at scale. Specifically, we propose to replace the non-smooth activation function ReLU with its smooth approximations (like Softplus or SILU) for ensuring architectural smoothness in adversarial training. Applying smooth activation functions in adversarial training improves adversarial robustness without sacrificing accuracy or incurring additional computation cost. Extensive experiments on ImageNet demonstrate the general effectiveness of adversarial training with smooth activation functions. By pushing the network scale to the very large EfficientNet-L1, adversarial training with smooth activation functions reports the state-of-the-art adversarial robustness on ImageNet, which substantially outperforms the prior art (Qin et al., 2019) by 9.5% for accuracy and 11.6% for robustness. Our results also corroborate the recent findings that there exist certain network architectures which have better adversarial robustness (Cubuk et al., 2018; Chen et al., 2020; Guo et al., 2020; Xie & et al., 2019; Huang et al., 2021). We hope these works together can encourage more researchers to investigate this direction.

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

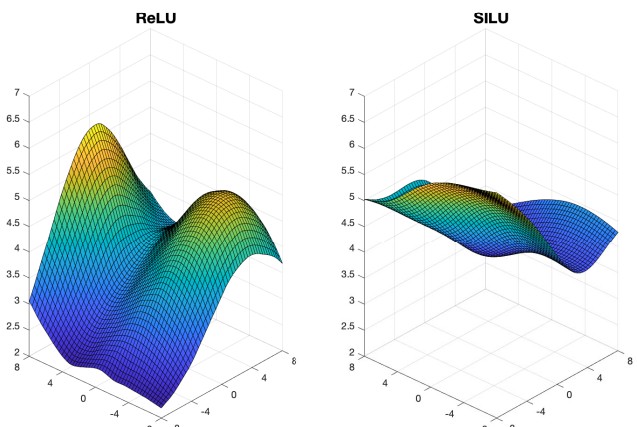

**Figure 5:** Comparison of loss landscapes between the network using ReLU and the network using SILU, on a randomly selected ImageNet sample.

## A   SANITY TESTS FOR ROBUSTNESS EVALUATION

Correctly performing robustness evaluation is non-trivial, as there exist many factors (*e.g.*, gradient mask) which may derail the adversarial attacker from accurately accessing the model performance. To avoid these evaluation pitfalls, we run a set of sanity tests, following the recommendations in (Carlini et al., 2019). Specifically, we take the ResNet-50 with SILU as the evaluation target.

**Robustness *vs*. attack iterations.** By increasing the attack iterations from 5 to 200, the resulted PGD attacker consistently hurts the model more. For example, by evaluating against PGD-{5, 10, 50}, the model reports the robustness of 48.7%, 43.7% and 42.7%, respectively. This evaluation finally gets converged at 42.3% when using PGD-200.

**Robustness *vs*. perturbation sizes $\epsilon$.** We also confirm a larger perturbation budget strengthens the attacker. By increasing $\epsilon$ from 4 to 8, the robustness drops more than 25%; the model will be completely circumvented if we set $\epsilon = 16$.

**Landscape visualization.** We compare the loss landscapes between ours and the ReLU baseline, on a randomly selected samples from ImageNet val set. Specifically, following (Li et al., 2018), the x/y-axis refer to the directions of adversarial perturbation, and z-axis refers to the corresponding cross-entropy loss. As shown in Figure 5, compared to the ReLU baseline, we observe that adversarial training with smooth activation functions produces a much smoother loss landscape. Note this smooth loss landscape can also be observed when using other smooth activation functions (besides SILU) and on other randomly selected images.

In summary, these observations confirm our robustness evaluation with PGD attacker is properly done in this paper.

## B   CIFAR-10 RESULTS

Though the main focus of this paper is to study adversarial robustness on the large-scale ImageNet dataset, in this section, we briefly check how adversarial training with smooth activation functions performs on CIFAR-10. We adversarially train ResNet-18 with the following settings: during training, the attacker is PGD-1 with maximum perturbation $\epsilon = 8$ and step size $\beta = 8$; we use AutoAttack (Croce & Hein, 2020) to holistically evaluate these models.

Firstly, as expected, we note Softplus, GELU and SmoothReLU all lead to better results than the ReLU baseline—compared to ReLU, all these three activation functions maintain a similar accuracy, but a much stronger robustness. For examples, the improvements on robustness are ranging from 0.7% (by GELU) to 2.3% (by Softplus).

However, unlike our ImageNet experiments where smooth activation functions demonstrate consistent improvements over ReLU, we note there are two exceptions on CIFAR-10—compared to ReLU, ELU and SILU even significantly hurt adversarial training. More interestingly, this inferior

performance can also be observed on the training set. For example, compared to ReLU, SILU yields 4.1% higher training error and ELU yields 10.6% higher training error. As suggested in (Pang et al., 2021; Wong et al., 2020), adversarial training on CIFAR-10 is highly sensitive to different parameter settings, therefore we leave the exploring of the "optimal" adversarial training setting with ELU and SILU on CIFAR-10 as a future work.

