# OpenReview forum: "Revisiting Activation Function Design for Improving Adversarial Robustness at Scale"
_ICLR.cc/2023/Conference — Submitted to ICLR 2023_

### Official Review · Reviewer_yesB · 2022-10-13

**Confidence:** 3
**Correctness:** 4
**Technical Novelty And Significance:** 3
**Empirical Novelty And Significance:** 4
**Recommendation:** 5

**Clarity, Quality, Novelty And Reproducibility:**


Clarity: the paper is well written and easy to understand.

Quality: the quality of the paper is good.

Novelty: as mentioned in the minor concern, some existing studies recognize smoothness as a problem. The authors of this submission need to provide some more insights to explain why it is different from the existing studies.

Reproducibility: I could not reproduce ImageNet data set in adversarial training because I do not have enough computation resources. Based on the Github https://github.com/locuslab/robust_overfitting of paper

Rice, Leslie, Eric Wong, and Zico Kolter. "Overfitting in adversarially robust deep learning." International Conference on Machine Learning. PMLR, 2020.

It takes days to run adversarial training in CIFAR-10 data set.



**Strength And Weaknesses:**


Strength:

The authors have a thorough experimental analysis in the ImageNet data set.

Weakness:

I have several concerns of this paper:

[1] My main concern is that the experiments are mainly for ImageNet, but no sufficient studies for other data sets, e.g., CIFAR-10. In Section B, it is mentioned that the performance of some of the smoothed activation functions are not as good as ReLU. The authors provides some explanation on the sensitivity of CIFAR-10 data set, but we can not claim that ImageNet is free from the same issue. As a result, some more in-depth analysis are wanted.

In addition, in the CIFAR-10 results, the authors only mention "the improvements on robustness are ranging from 0.7% (by GELU) to 2.3% (by Softplus)". In Table 2 of Croce & Hein (2022), there is a long list of CIFAR-10 results from different papers. The authors of this submission may provide some more details about the performance.

The paper title has the word "at scale", but it is not clear why the smoothing techniques cannot benefit small scale data sets like CIFAR-10. I would suggest the authors to try different training size to clarify whether the word "at scale" is really important or not. If using 50000 ImageNet samples from 10 classes, will it have the same problem as CIFAR-10? On the other hand, in literature such as

Gowal, Sven, et al. "Improving robustness using generated data." Advances in Neural Information Processing Systems 34 (2021): 4218-4233.

and

Carmon, Yair, et al. "Unlabeled data improves adversarial robustness." Advances in Neural Information Processing Systems 32 (2019).

it is observed that more data improves adversarial training for CIFAR-10. If we use the 1M data provided by Gowal et. al. (2021, https://github.com/deepmind/deepmind-research/tree/master/adversarial_robustness), will we observe similar observation as the ImageNet data set?


[2] All papers in the reference list are published before 2021. Could the authors supplement the introduction with some more updated literature in the recent two years?


Minor issue:

[1] Some theory literature already consider some smoothness problem in adversarial training. For example, in

Allen-Zhu, Zeyuan, and Yuanzhi Li. "Feature purification: How adversarial training performs robust deep learning." 2021 IEEE 62nd Annual Symposium on Foundations of Computer Science (FOCS). IEEE, 2022.

To avoid the non-smooth issue, they directly drop the attack if the gradient is small (the equation above Definition 4.3). It mentions that "because otherwise A(f, x, y) is not Lipscthiz continuous at those points." in footnote 9 (https://arxiv.org/pdf/2005.10190.pdf).

Could you help explain what are the difference in the intuitions between the submission and this paper?



**Summary Of The Paper:**

The paper provides a lot of ImageNet experiments justifying that smooth activation function helps adversarial training.

**Summary Of The Review:**

The authors provide a lot of experiments in ImageNet data set to show that adversarial training benefits from smoothed activation functions. I would encourage the authors to (1) provide detailed experiments in other data sets, e.g., CIFAR-10, and show that the proposed methods are uniformly good for all (or most) data sets; (2) update the literature review with recent studies and emphasize the novelties.

I still rate the submission as 5, because the experimental results for ImageNet data set are thorough, and no previous empirical study has done as good as this paper based on my knowledge.

---

### Official Review · Reviewer_qQzz · 2022-10-14

**Confidence:** 4
**Correctness:** 3
**Technical Novelty And Significance:** 2
**Empirical Novelty And Significance:** 3
**Recommendation:** 5

**Clarity, Quality, Novelty And Reproducibility:**

The paper is mostly clear and well-written (aside from a few typos). A focused analysis of the effect of activation function smoothness on adversarial robustness is novel to the best of my knowledge. Reproducibility is ensured through the authors' code release.

I believe the authors are somewhat imprecise concerning terminology. "Gradient quality" is repeatedly mentioned, but is never defined. I believe the authors should provide a more formal definition (e.g., Lipschitzness of the gradient) or settle for a less ambiguous term.  Analogously, concerning section 3, I think it is somewhat imprecise to state that a different activation function is being employed during the backward pass. It would be more precise to say that the activation function gradient is replaced by a continuous approximation.

**Strength And Weaknesses:**

**Strengths**.
Analyzing the interplay between activation function smoothness and adversarial robustness is definitely a topic of interest to the community. If confirmed, the provided experimental results could push practitioners towards the use of smooth activation functions, as these would appear to yields better empirical robustness at no cost in standard accuracy or training cost.

**Weaknesses**.
As the authors themselves acknowledge in appendix A, "*correctly performing robustness evaluation is non-trivial*". In this sense, while the appendix itself provides some sanity checks, I believe these are insufficient to ensure the validity of the claims.
- the vast majority of the training experiments relies on a single-step PGD attack (except for a small experiment in section 3.3): would the claims still hold when running the attacks for longer?
- it would appear to me that the PGD attack step size is never tuned, neither at training nor at evaluation. Could the authors provide more information regarding this? The attack step size might be fairly sensitive to the activation function.
- the evaluation of empirical robustness is inherently tied to the employed attack. The authors are focussing on PGD, but would the claims still hold under different attack methods? For instance, black-box attack schemes (e.g., Square Attack), or AutoAttack.
- the evaluation mostly focuses on a single dataset (ImageNet). CIFAR-10 results (in the appendix) are only drafted, and seem to radically differ from those presented in the main paper. It would be nice to have evidence that the claims hold beyond a single dataset.
- no guarantee about the actual robustness of a network (empirical robustness might always be an artifact of the attack itself) can be given without formal verification techniques. It would be very interesting to check whether the verified adversarial robustness of the trained networks (a bound can be computed by running [IBP](https://arxiv.org/abs/1810.12715) or adapting [CROWN](https://arxiv.org/abs/1811.00866) for the employed activation functions) is affected by the use of smooth activations.

I would also like to point out that formal verification techniques usually rely on branch and bound (see [VNN-COMP-21](https://arxiv.org/abs/2109.00498)), which is guaranteed to terminate in finite time only for piece-wise linear activations. Therefore, verifiability of adversarial robustness is likely to be negatively affected by the use of smooth activation functions.
It would be nice if the authors could mention this in the introduction or related work.


**Summary Of The Paper:**

The paper presents an experimental evaluation on the use of smooth activation functions, approximating ReLU, in the context of adversarial robustness on ImageNet. The results would appear to show that smooth activations increase empirical adversarial robustness without negatively affecting standard accuracy. Furthermore, the experiments confirm the belief that capacity is crucial to enhance trade-offs between standard and robust accuracy.

**Summary Of The Review:**

While the subject of this experimental paper is interesting and novel, I believe the authors' claims are not adequately supported due to a series of weaknesses of the experimental evaluation.
I am more than willing to increase my score if the authors address the aforementioned limitations.

---

### Official Review · Reviewer_zKRu · 2022-10-19

**Confidence:** 5
**Correctness:** 2
**Technical Novelty And Significance:** 1
**Empirical Novelty And Significance:** 2
**Recommendation:** 3

**Clarity, Quality, Novelty And Reproducibility:**

Clarity and Quality: In general, this paper is well-written and easy to follow. Figures and Tables make the paper more readable.

Novelty: The novelty of this work is not high. As it is not new that smooth activation function is beneficial to model robustness (such as this cvpr paper: https://openaccess.thecvf.com/content/ICCV2021/papers/Singla_Low_Curvature_Activations_Reduce_Overfitting_in_Adversarial_Training_ICCV_2021_paper.pdf)

Reproducibility: The code is provided anonymously, which is helpful for reproducing the results.

**Strength And Weaknesses:**

Strength:

This paper comprehensively study the effect of smooth activation function from different perspectives by extensive experiments.

Weakness:

1. To comprehensively evaluate the robustness of a model, it is not enough to just increase the iteration number of PGD. The authors should also try other methods, such as the ones using loss functions other than cross-entropy and some margin based attack methods (like FAB: https://arxiv.org/abs/1907.02044). Currently, the most reliable method to evaluate robustness is AutoAttack (http://proceedings.mlr.press/v119/croce20b/croce20b.pdf), the authors should report the robust accuracy on AutoAttack, which is more accurate than PGD.

2. Using stronger adversarial attack does not necessarily lead to more robust models. For example, 1-step fast adversarial training can still achieve competitive performance, while fitting some out-of-distribution training instances with strong adversarial perturbations may hurt the performance. (https://arxiv.org/abs/1910.08051) Therefore, some related explanations in Section 3.2 are not convincing.

3. This is a pure empirical work, as the difference between some methods is not very big, the authors should run the experiments multiple times and report the performance variance.

4. In section 3.2, smooth activation functions are used only for backward, the authors should explain why.

5. Just one setting (such as model architecture, the dataset) is studied in Section 3.3, so the conclusions in this section are not convincing.



**Summary Of The Paper:**

This paper studies the effects of smooth activation functions on model robustness. It claims smooth activation functions lead to better performance compared with ReLU. The authors conduct extensive experiments on ImageNet to illustrate how smooth activation functions affect the adversarial example generation and model parameter update. In addition, this work also studies how the model capacity and input resolution affect the performance of models with smooth activation function.

**Summary Of The Review:**

Based on the concerns and weakness pointed above, I cannot recommend this paper to appear in ICLR. I welcome the authors to address my concerns during the rebuttal, after which I will do another evaluation.

---

### Official Review · Reviewer_U7PC · 2022-10-25

**Confidence:** 4
**Correctness:** 3
**Technical Novelty And Significance:** 2
**Empirical Novelty And Significance:** 2
**Recommendation:** 3

**Clarity, Quality, Novelty And Reproducibility:**

Clarity, Quality, Novelty And Reproducibility are provided in Strength and Weakness Section.

**Strength And Weaknesses:**


Strength
- This paper focuses on ImageNet dataset. To my best knoweldge, it is the first time to do extensive experiments for the large scale ImageNet dataset.
- This paper proposes a new algorithm for adversarial training with ReLU in forward pass and with smooth activation in backward pass without additional computational cost. It improves the adversarial robustness without hampering standard accuracy.
- This paper empirically shows SiLU is the most effective in adversarial robustness among various smooth activation functions on ImageNet.

Weakness
- This paper lacks intuitive explanation (and theoretical studies) about why smooth activation functions improve the adversarial robustness.
- The experiments are conducted on only ImageNet. It is necessary to conduct experiments with other benchmark datasets to support the main claim. If the results of other benchmark datasets are not consistent, the novelty would be small.
- The evaluation is not properly executed. Checking for gradient masking is very crucial. I admit that autoattack is difficult to use on ImageNet dataset. At least, one black-box attack such as Square Attack [1] is necessary for checking the gradient masking.

[1] Andriushchenko, Maksym and Croce, Francesco and Flammarion, Nicolas and Hein, Matthias, Square Attack: a query-efficient black-box adversarial attack via random search, In ECCV, 2020.

Question :
- Authors insist that the gradient from a smooth activation function is better than ReLU. In what sense is it a better gradient?

Minor
[At second paragraph in 4th section] Noe that we keep => Note that we keep

**Summary Of The Paper:**

This paper finds that non-smooth ReLU activation function weaken the adversarial robustness and smooth activation functions such as SiLU can improve the adversarial robustness. The experiments are based on large-scale ImageNet dataset and show the state-of-the-art performance.

**Summary Of The Review:**

The authors empirically show that the smooth activation function improves the adversarial robustness on ImageNet dataset. Extensive experiments on ImageNet dataset are interesting. But, I have concerns about the checking for gradient masking and consistent results for other benchmark dataset and theoretical results ( or explanation ) of why smooth activation function improves adversarial robustness.

---

### Decision · Program_Chairs · 2023-01-20

**Decision:**

Reject

**Justification For Why Not Higher Score:**

See above for weaknesses

**Justification For Why Not Lower Score:**

I recommend rejection.

**Metareview: Summary, Strengths And Weaknesses:**

This paper studies the effect of smoothness of activations in robustness and finds that non-smooth ReLU activation function weaken the adversarial robustness and smooth activation functions such as SiLU can improve the adversarial robustness. The paper is mostly clear and well-written. However, there are two critical weaknesses pointed by reviewers: one is the lack of extensive evaluation using adaptive attacks and studying the potential issue of gradient masking; the other is the lack of intuitive explanation (and theoretical studies) about why smooth activation functions. Given all, I think the paper needs a bit more work to be accepted.